# IceCache: Memory-efficient KV-cache Management for Long-Sequence LLMs

**Yuzhen Mao**
Simon Fraser University, BC, Canada
`yuzhenm@sfu.ca`

**Qitong Wang**
Harvard University, Cambridge, USA
`qitong@seas.harvard.edu`

**Martin Ester**
Simon Fraser University, BC, Canada
`ester@sfu.ca`

**Ke Li**
Simon Fraser University, BC, Canada
`keli@sfu.ca`

## Abstract

Key-Value (KV) cache plays a crucial role in accelerating inference in large language models (LLMs) by storing intermediate attention states and avoiding redundant computation during autoregressive generation. However, its memory footprint scales linearly with sequence length, often leading to severe memory bottlenecks on resource-constrained hardware. Prior work has explored offloading KV-cache to the CPU while retaining only a subset on the GPU, but these approaches often rely on imprecise token selection and suffer performance degradation in long-generation tasks such as chain-of-thought reasoning. In this paper, we propose a novel KV-cache management strategy, IceCache, which integrates semantic token clustering with PagedAttention. By organizing semantically related tokens into contiguous memory regions managed by a hierarchical, dynamically updatable data structure, our method enables more efficient token selection and better utilization of memory bandwidth during CPU–GPU transfers. Experimental results on Long-Bench show that, with a 256-token budget, IceCache maintains 99% of the original accuracy achieved by the full KV-cache model. Moreover, compared to other offloading-based methods, IceCache attains competitive or even superior latency and accuracy while using only 25% of the KV-cache token budget, demonstrating its effectiveness in long-sequence scenarios. The code is available on our project website at `https://yuzhenmao.github.io/IceCache/`.

## 1 Introduction

Key-Value (KV) cache is a critical component in modern large language models (LLMs), which stores the intermediate attention outputs for each token, allowing the model to reuse these computations in subsequent forward passes. This mechanism is particularly important for autoregressive generation, where tokens are produced sequentially. However, a fundamental challenge of KV-cache lies in its memory consumption: as the generated sequence length increases, the cache size grows linearly, often leading to severe memory pressure or out-of-memory errors on devices with limited resources.

Recent studies (Zhang et al., 2024b; Tang et al., 2024; Xiao et al., 2023) have shown that, despite the growing size of the KV-cache, only a small subset of tokens contributes disproportionately to generation accuracy. Building on this insight, subsequent work (Chen et al., 2024a; Lee et al., 2024; Chen et al., 2024b) offloads the KV-cache to the CPU while dynamically retaining only the most important entries on the GPU. However, many existing approaches lack precise mechanisms for identifying truly relevant tokens, resulting in low hit rates for the most relevant cache entries. In addition, they often

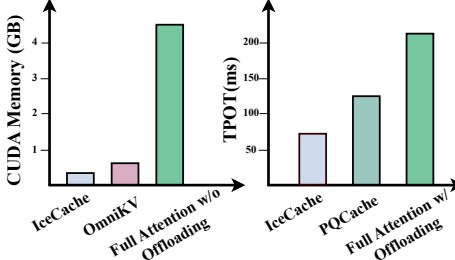

Figure 1: IceCache has the best trade-off between CUDA memory footprint and time-per-output-token (TPOT) on A100 at a 36k sequence length. Baselines are chosen to represent high-accuracy (Left) and memory-efficient (Right) methods.

lack effective update mechanisms to manage the KV-cache pool as new tokens are generated during decoding. Therefore, in scenarios involving long-generation tasks, such as long-context summarization, multi-step reasoning, and chain-of-thought (CoT) generation, previous methods experience significant performance degradation (Li et al., 2024a).

To improve the identification of relevant KV-cache entries, we propose an innovative approach, IceCache, which integrates semantic token clustering with PagedAttention (Kwon et al., 2023), a widely adopted memory management technique that stores KV-cache entries in non-contiguous pages. As illustrated in Figure 2, by grouping semantically related tokens into the same memory pages, our method increases the likelihood that relevant tokens are co-located, thereby improving page selection hit rates and enhancing memory bandwidth utilization during CPU–GPU transfers. Furthermore, IceCache employs a hierarchical data structure that can be efficiently updated during decoding, mitigating the performance degradation commonly observed in long-generation tasks. As a result, IceCache enables more effective and adaptive KV-cache management – it achieves superior latency and accuracy compared to state-of-the-art baselines under the same KV-cache budget, and still attains comparable accuracy while using substantially smaller budgets.

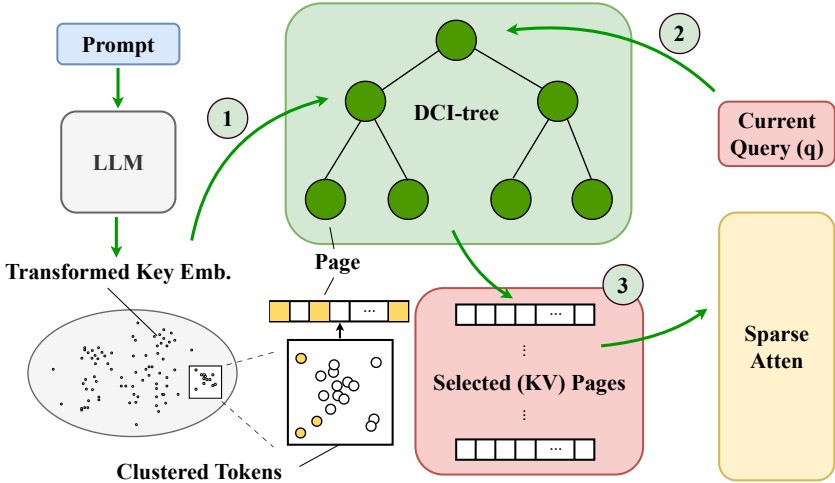

Figure 2: Illustration of IceCache. (1) During the prefill stage, tokens are indexed into a hierarchical data structure (the DCI-tree) according to their semantic similarity in the transformed key-embedding space. Each leaf node of the DCI-tree corresponds to a physical memory page. (2) During the decoding stage, given a query $q$, IceCache performs a tree search to identify the top-$k$ tokens most relevant to $q$. The zoomed-in section at the bottom illustrates that these relevant tokens (highlighted in yellow) tend to be clustered within the same leaf nodes and are stored together in corresponding memory pages. (3) After the query-aware retrieval, the pages containing the relevant tokens are selected, and all tokens within these pages are used in the subsequent sparse attention with $q$.

In summary, IceCache makes the following contributions:

- We propose a KV-cache offloading strategy that substantially improves retrieval hit rates by grouping semantically similar tokens into the same memory pages.
- We develop a multi-level DCI algorithm that supports efficient indexing, fast retrieval, and dynamic updates, enabling effective maintenance of the KV-cache data structure.
- We design a pipelining scheme that overlaps CPU and GPU computations, effectively hiding the latency associated with indexing and retrieval.

We evaluated IceCache under constrained GPU memory budgets on the Passkey Retrieval (Mohtashami & Jaggi, 2023), LongBench (Bai et al., 2023), and GSM8K Chain-of-Thought (CoT) reasoning (Wei et al., 2022) using four popular open-source LLMs: Llama3.1-8B-Instruct, Mistral-7B-Instruct-v0.2, LongChat-7B-v1.5 and Qwen3-32B. Across diverse tasks, including open-domain QA, multi-hop reasoning, academic reading comprehension, long-context summarization and long-context generation, IceCache consistently outperformed six state-of-the-art KV-cache baselines.

Notably, IceCache sustained near-oracle performance on most tasks using only a small fraction (as small as 64 tokens) of the original KV-cache size. Furthermore, by leveraging its hierarchical and dynamically updatable data structure, IceCache effectively mitigates performance degradation in long-generation tasks. As shown in Figure 1, the empirical results demonstrate that IceCache establishes a new state-of-the-art in accuracy–efficiency trade-offs, achieving superior accuracy while significantly reducing CUDA memory usage and decoding latency compared to all existing baselines.

## 2 RELATED WORK

### 2.1 KV-CACHE MANAGEMENT VIA EVICTION AND OFFLOADING

Existing approaches to KV-cache management generally fall into two categories: eviction-based methods and offloading-based methods. Eviction-based methods are typically faster, as they avoid data transfer between the GPU and the CPU. However, they often rely on static selection strategies and fail to adapt to changing contexts. In contrast, offloading-based methods aim to preserve important KV-cache entries more accurately by dynamically transferring data between GPU and CPU, but introduce additional latency.

Eviction-based approaches include H2O (Zhang et al., 2024b), which retains only a subset of tokens selected based on attention scores; StreamingLLM (Xiao et al., 2023), which preserves sink tokens and recent tokens to approximate attention; and SnapKV (Li et al., 2024b), which selects key embeddings from the tail of the prompt to guide subsequent decoding. While efficient, these methods may struggle to maintain accuracy in long-generation scenarios due to their limited flexibility in adapting to evolving contexts. Offloading-based methods dynamically manage KV-cache storage across devices. MagicPiG (Chen et al., 2024b) uses sampling techniques based on Locality Sensitive Hashing (LSH) to approximate attention. OmniKV (Hao et al., 2025) improves efficiency by reusing important tokens across consecutive layers. PQCache (Zhang et al., 2024a) applies product quantization to compress the KV-cache and approximate attention computation. Although these approaches better preserve critical information, they incur additional communication costs from GPU–CPU transfers.

### 2.2 PAGED KV-CACHE MANAGEMENT

PagedAttention (Kwon et al., 2023) is an innovative memory management technique designed to optimize the KV-cache of LLMs. It addresses the challenges by introducing a paging mechanism similar to virtual memory systems in operating systems. This approach divides the KV-cache into fixed-size pages, allowing for more efficient memory allocation and management. By doing so, PagedAttention improves GPU memory utilization by reducing fragmentation and enabling longer context windows without sacrificing performance. However, unlike the methods discussed in the previous subsection, PagedAttention does not address the fundamental issue that the KV-cache continuously grows during decoding.

Quest (Tang et al., 2024) and ArkVale (Chen et al., 2024a) are two query-aware criticality estimation algorithms built on PagedAttention. They effectively identify relevant KV-cache tokens and perform self-attention selectively on the chosen tokens. For each page, Quest and ArkVale calculate an upper bound using the feature values of the Key vector for each page's criticality estimation. Given all criticality scores of the pages, Top-$k$ pages are chosen to perform approximate self-attention, where $k$ is a preset constant (e.g. 128, 256). Additionally, ArkVale integrates the GPU-CPU offloading into the system to further save GPU memory. However, a key limitation of both Quest and ArkVale is that they construct memory pages according to the original token order. As a result, tokens that are semantically relevant to a given query may be scattered across multiple pages. In contrast, IceCache performs semantic clustering during indexing, grouping similar tokens into the same pages. This organization increases the likelihood that relevant tokens are co-located, improving retrieval hit rates and reducing unnecessary memory transfers.

## 3 BACKGROUND

### 3.1 ATTENTION MECHANISM AND SPARSE ATTENTION

Mathematically, the attention operation takes three matrices as input, $\mathbf{K} \in \mathbb{R}^{m \times d}, \mathbf{Q} \in \mathbb{R}^{n \times d}, \mathbf{V} \in \mathbb{R}^{m \times d'}$, which denote keys, queries and values, respectively. Optionally, it may also take in a mask as input, $\mathbf{S} \in \mathbb{R}^{n \times m}$, whose entries are either 0 or 1. The $i$th rows of $\mathbf{K}$, $\mathbf{Q}$ and $\mathbf{V}$, denoted as $\mathbf{k}_i$, $\mathbf{q}_i$ and $\mathbf{v}_i$, represent the $i$th key, query, and value, respectively. The entry of $\mathbf{S}$ in the $i$th row and $j$th

column, denoted as $s_{i,j}$, represents whether the $i$th query is allowed to attend to the $j$th key — if it is 1, it would be allowed; if it is 0, it would not be. A common masking scheme is the causal mask, where $s_{i,j}$ is 1 if $i \geq j$ and 0 otherwise. Keys and queries have the same dimension $d$, and each key is associated with a value, and so the number of keys and values is the same and denoted as $m$. The attention operation computes the attention weight matrix $\mathbf{A} \in \mathbb{R}^{n \times m}$. Its entry in the $i$th row and $j$th column, denoted as $a_{i,j}$, is computed with the following formula:

$$a_{i,j} = \frac{s_{i,j} \exp\left(\frac{\mathbf{q}_i^\top \mathbf{k}_j}{\sqrt{d}}\right)}{\sum_{j'=1}^{m} s_{i,j'} \exp\left(\frac{\mathbf{q}_i^\top \mathbf{k}_{j'}}{\sqrt{d}}\right)} \tag{1}$$

The attention matrix $\mathbf{A}$ is typically sparse (Nikita et al., 2020; Gupta et al., 2021), i.e., in each row of $\mathbf{A}$, only a few attention weights have significant (large) values, while the majority of the remaining values are close to zero. If we can somehow identify the $k$ unmasked keys that receive the highest attention weights for each query $\mathbf{q}_i$ without computing the attention weights for all keys, the original attention matrix $\mathbf{A}$ can be approximated by only computing the inner product for the identified keys, which can save a significant amount of computational resources.

## 3.2 LLM INFERENCE

The inference process of LLMs primarily comprises two key stages: the prefill (or prompt) stage and the decoding (or generation) stage. In the prefill stage, the model takes an input prompt sequence of length $s_{in}$ and processes it through all layers of the LLM. During this process, the keys and values for each token in the sequence are computed and stored as part of the KV-cache. The decoding stage begins once the prompt has been processed. Here, the model generates output tokens one step at a time, using and updating the KV-cache iteratively. For each decoding step, the current token's computation depends on the stored keys and values from previous tokens, allowing the model to maintain context over the sequence. The KV-cache thus plays a crucial role in enabling efficient autoregressive generation by reducing redundant computations and maintaining information about past tokens.

## 3.3 MULTI-LEVEL DCI

**Prioritized Dynamic Continuous Indexing (P-DCI).** Li & Malik (2016) proposed an exact, randomized algorithm designed to perform efficient $k$-nearest neighbour (k-NN) searches in high-dimensional spaces. Unlike traditional methods, DCI avoids space partitioning, which scales poorly to high dimensions. It does so by constructing multiple indices, each of which allows for points to be efficiently ranked by a lower bound on the distances between them and the query. The lower bound is formed by projecting the distances along a random vector associated with each index. P-DCI (Li & Malik, 2017) aggregates over the lower bounds given by each index to produce a tighter lower bound and uses a priority queue to order points so that points with smaller lower bounds are processed earlier than points with larger lower bounds. This approach significantly reduces the number of distance evaluations and memory usage compared to methods like Locality-Sensitive Hashing (LSH). Mao et al. (2024) first introduced P-DCI into sparse attention mechanisms to accelerate LLM inference.

**Multi-level Dynamic Continuous Indexing (M-DCI).** In this work, we extend P-DCI by introducing a hierarchical data structure to further enhance search efficiency. The index is organized into multiple levels, each containing a subset of data points. Points are randomly promoted to higher levels, forming a hierarchical data structure. Each point at a lower level is assigned a parent in the next higher level, typically the nearest neighbour among the promoted points. This creates "nodes" or clusters of points sharing the same parent. When querying, the algorithm starts at the top level, using P-DCI to find the $k$-closest points to the query. It then recursively searches within the nodes associated with these points at the next lower level, continuing this process down the hierarchy. This multi-level approach allows M-DCI to focus computational resources on the most promising regions of the index, effectively narrowing down the search space and improving query times, especially in datasets with high intrinsic dimensionality.

## 4 ICECACHE

We propose an innovative approach, named IceCache, that integrates token clustering with KV-cache storage. Our method consists of three steps: (1) indexing; (2) page selection; and (3) bulk loading. The indexing step occurs either during the prompt processing phase—when IceCache constructs a

hierarchical data structure, referred to as the DCI-tree, for the prompt key embeddings; or when new window pages are offloaded to the CPU. In this step, similar tokens are grouped into units called nodes, rather than being stored sequentially in virtual memory pages as in PagedAttention. Here, a node denotes a group of data points that share the same parent in the tree hierarchy. The next two steps take place during the token generation (decoding) phase. In the page selection step, IceCache employs a fast Approximate Nearest Neighbor (ANN) search algorithm, M-DCI, to independently select the top-$k$ most relevant pages for each attention head. Finally, in the bulk loading step, the selected pages are efficiently transferred from the CPU back to the GPU. IceCache also overlaps the indexing (a CPU-intensive operation) with ongoing GPU computations to further reduce the latency.

We provide further details on each of these three steps in the following subsections and illustrate the method in Fig 3.

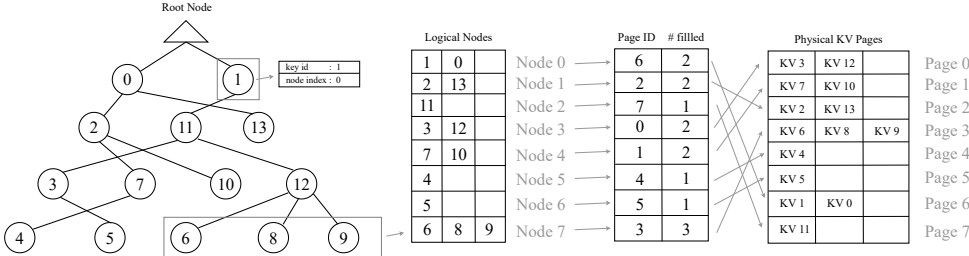

Figure 3: Illustration of DCI-tree and IceCache: The hierarchical data structure on the left visualizes the result of indexing key embeddings, DCI-tree, where each tree node stores metadata for the tokens, such as the key ID and node index. The tables on the right depict the mapping between nodes in the DCI-tree and the corresponding pages in physical memory. For each selected node, a mapping table is used to locate the memory region containing the associated key-value embeddings.

### 4.1 INDEXING: CLUSTERING KEY EMBEDDINGS INTO A HIERARCHICAL TREE

During decoding, the dominant bottleneck is often not the computation but memory bandwidth. As the KV-cache grows with sequence length, each decoding step requires repeatedly loading large amounts of KV embeddings from GPU memory. This leads to memory-bandwidth saturation, significantly degrading latency scalability. PagedAttention (Kwon et al., 2023) mitigates this issue by introducing a virtual memory abstraction for the KV-cache. Instead of storing the KV-cache as a single contiguous tensor, it organizes KVs into fixed-size memory pages and maintains a page table to map logical token positions to physical memory locations. Within each page, consecutive tokens are stored contiguously, reducing fragmentation. This paging mechanism enables more flexible memory allocation and more efficient KV-cache management during decoding.

Building on this design, subsequent KV-cache optimization methods such as Quest (Tang et al., 2024) and ArkVale (Chen et al., 2024a) leverage PagedAttention's paging abstraction while incorporating query-aware page selection. These approaches estimate the importance of each page during decoding and utilize only the top-$k$ pages to approximate attention more efficiently.

However, since pages are constructed based on the original token order, tokens relevant to a given query are often distributed across multiple pages. Therefore, retrieving them requires loading entire pages that may contain many irrelevant tokens, leading to unnecessary memory overhead and reduced selection precision. In contrast, IceCache organizes KV-cache entries based on semantic similarity, concentrating relevant tokens within fewer pages. This design improves retrieval efficiency and achieves superior performance while reducing the number of pages accessed.

Specifically, IceCache organizes key-value embeddings into memory pages through a fundamentally different indexing strategy: instead of relying on the token's original order, IceCache constructs a separate hierarchical tree data structure for each attention head, called a DCI-tree, which clusters tokens based on the semantic similarity of their key embeddings. Each node in the DCI-tree represents a small group of semantically related tokens that share a common parent, effectively forming a localized cluster. From a memory system perspective, IceCache maps each node directly to a memory page, thereby preserving semantic locality in storage and enabling efficient access during

decoding. By clustering semantically similar tokens into the same nodes/pages, IceCache enables more targeted and efficient retrieval during decoding.

Moreover, the DCI-tree data structure used by IceCache is designed for efficient incremental updates. As new windows of tokens (e.g., from a sliding window in long-context scenarios) are offloaded to the CPU, each token is inserted into the appropriate node in the DCI-tree based on its key embedding. When a node exceeds the maximum page size, new pages are dynamically allocated to maintain balance. This adaptive tree maintenance ensures that the index remains both semantically meaningful and efficient, making IceCache particularly effective for long-sequence generation.

In summary, while prior methods treat KV-cache page layout as a static memory allocation problem, IceCache introduces a dynamic, semantically-aware data structure that preserves key similarity across time. This enables more focused page retrieval, reduces memory fragmentation, and supports more efficient decoding. Furthermore, by performing indexing during the prompt or CPU offloading phase, IceCache amortizes the tree construction cost and avoids incurring additional latency during inference.

## 4.2 PAGE SELECTION: HEAD-SPECIFIC ANN SEARCH WITH FINE-GRAINED RETRIEVAL

During the decoding phase, given a query embedding, IceCache performs head-specific page selection to identify the most relevant pages for each attention head. Leveraging the hierarchical DCI-tree constructed during indexing, we apply the fast approximate nearest neighbor (ANN) search algorithm described in Section 3.3, M-DCI, to retrieve the top-$k$ pages that are important to the current query for each head independently. For group-query attention (GQA), where multiple query heads share the same set of key embeddings, we compute the union of the selected pages across queries within the same group and share the resulting pages among them, following Yuan et al. (2025).

The detailed pseudo-code of this procedure is provided in Appendix B.2.

## 4.3 BULK LOADING AND PIPELINING

In this section, we present how we optimize the efficiency of the IceCache workflow, using bulk loading and pipelining. Our key observation is that the selected pages are not contiguous in either main memory or GPU memory. As a result, individual transfer of these pages between main memory and GPU memory is highly inefficient. To overcome this issue, we designed bulk loading algorithms to efficiently offload and backload the selected pages, using two CPU and GPU backload buffers. In addition, we carefully designed an efficient pipeline of prefilling calculations, KV-cache offloading, and DCI indexing, for efficient prefilling. We illustrate our bulk back-loading workflow in Figure 4.

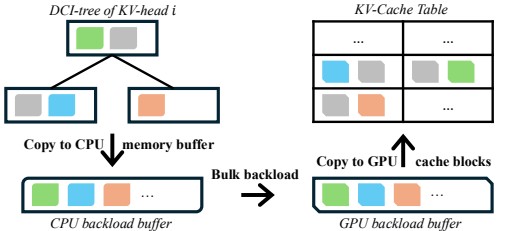
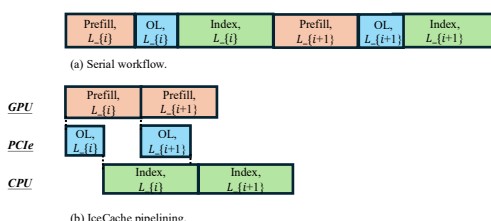

Figure 4: After IceCache selects important KV-pages, it aggregates all selected pages into a contiguous CPU preloading buffer. This buffer is then transferred via high-throughput PCIe transaction to a pre-allocated GPU buffer. Finally, the transferred blocks are scattered into their exact locations in the KV-Cache table. This bulk transfer avoids many small copying operations over PCIe and significantly improves utilization.

Figure 5: (a) Baseline serial workflow, where prefilling, offloading (OL), and indexing are executed strictly in sequence. (b) IceCache pipelining, where GPU prefilling overlaps with KV-offloading via PCIe and CPU-side DCI indexing. Once KVs of layer $i$ ($L_i$) arrive in CPU memory, Li-DCI-tree indexing progresses in parallel with GPU prefilling and offloading of the subsequent layer ($L_{i+1}$). This results in significantly reduced end-to-end prefilling latency.

Offloading follows a similar procedure, except that the steps are executed in reverse. After identifying the most relevant pages (indicated by color), we filter out those already resident in GPU memory from previous token generations. The remaining pages are then aggregated into a pre-allocated CPU-

memory buffer, enabling a single high-throughput PCIe transfer into a pre-allocated GPU-memory buffer. Once the pages arrive in the GPU, we perform a scatter operation to load them into the appropriate slots of the KV-Cache table.

Figure 5(b) illustrates the pipelining of IceCache prefilling (other minor stages are omitted for clarity). After the KV embeddings are generated on the GPU, we simultaneously launch the prefilling calculation and the offloading transfer. The DCI index construction starts building the index along with the prefilling calculation, right after the KV states fully arrive in the main memory. This approach allows the offloading and indexing latencies to be largely hidden by the main prefilling computation. Furthermore, IceCache can be easily extended with page reuse techniques (Liu et al., 2023; Hao et al., 2025) to further accelerate prefilling.

## 5 Experiments

### 5.1 Settings

To comprehensively evaluate IceCache, we conduct experiments across models of varying scales and attention architectures. We first apply IceCache to Llama-3.1-8B-Instruct and Mistral-7B-Instruct-v0.2, two widely adopted open-source LLMs employing group-query attention (GQA) (Ainslie et al., 2023). To further assess scalability and architectural generality, we extend our evaluation to Qwen3-32B, a larger-scale model, and LongChat-7B-v1.5, which employs standard multi-head attention (Vaswani et al., 2017) rather than group-query attention. Our evaluation proceeds in four stages. We first measure recall in retrieving important tokens (Mohtashami & Jaggi, 2023). To evaluate the performance of IceCache on long generation tasks, we further conduct experiments on LongBench (Bai et al., 2023), which contains long generation tasks such as summarization and code generation, and GSM8K (Cobbe et al., 2021), which requires chain-of-thought reasoning. Finally, we examine its scalability on RULER (Hsieh et al., 2024) under extremely long-context settings.

Our experimental platform comprises an NVIDIA A100 40GB PCIe GPU (for small models) or an NVIDIA H100 80GB PCIe GPU (for large models). For all experiments, we fix the number of CPU threads to 64. The software stack includes CUDA version 12.2, PyTorch version 2.4.1, and HuggingFace Transformers version 4.57.1. We implement IceCache on top of HuggingFace Transformers, utilizing FlashInfer (Ye et al., 2025) for the attention kernel operation. As prior research indicates (Tang et al., 2024), the initial layers of the model exhibit relatively low sparsity. Therefore, neither IceCache nor baseline methods are applied to the first two layers of the models.

### 5.2 Passkey Retrieval Accuracy

We first evaluate IceCache's effectiveness in handling long-range dependencies using the passkey retrieval task. We consider context lengths from 10k words to 100k words, and test with the size of cache budget $= \{256, 128, 64\}$. For each length, 100 test cases are generated with passkeys inserted at various positions from 0% to 95% of the total context length in increments of 5%. The results are illustrated in Fig. 6. As shown in the figure, IceCache dynamically assesses the importance of evicted pages and recall crucial ones on demand, consistently maintaining 100% retrieval accuracy across all tested budget sizes.

### 5.3 LongBench Evaluation

To assess the performance of our method in long-context scenarios, we conduct a comprehensive evaluation on the LongBench benchmark (Bai et al., 2023). LongBench is designed to evaluate how well LLMs understand and reason over long-context inputs across diverse real-world tasks. We compare IceCache (ICE) against six state-of-the-art KV cache optimization methods, including MagicPig (MPG) (Chen et al., 2024b), ArkVale (AKV) Chen et al. (2024a), SnapKV (SKV) Li et al. (2024b), StreamingLLM (SLM) Tang et al. (2024), OmniKV (OKV) Hao et al. (2025) and PQCache (PQC) Zhang et al. (2024a) as baselines. We also include the results of full KV-cache (FULL) and ground-truth top-$k$ KV-cache (TOP-$k$) as baselines. Notably, OmniKV offloads only a subset of layers while keeping all remaining layers fully resident on the GPU, resulting in substantially higher memory usage than other methods. The detailed results are presented in Table 1.

#### 5.3.1 Accuracy Analysis

**Accuracy on Llama-3.1-8B-Instruct.** On Llama-3.1-8B, the effectiveness of IceCache is particularly pronounced. Most impressively, with a highly constrained KV-Cache budget of just 64, our method achieves an average accuracy of 47.8. This result alone surpasses the strongest baseline,

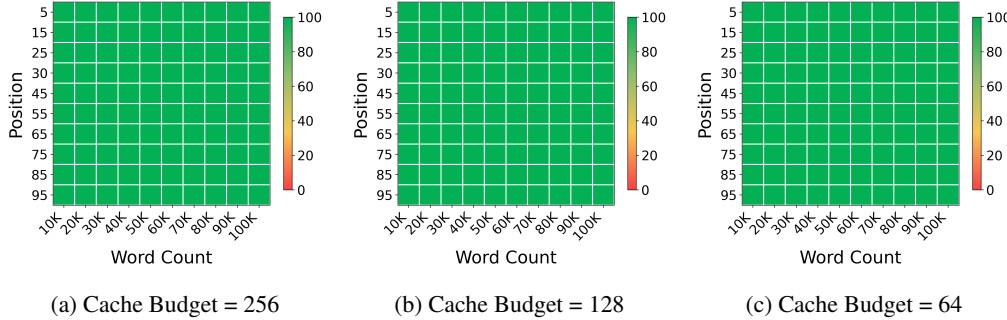

(a) Cache Budget = 256      (b) Cache Budget = 128      (c) Cache Budget = 64

Figure 6: Passkey retrieval accuracy of IceCache on Llama3.1-8B-Instruct. The horizontal axis indicates the relative insertion position (%) of the passkey, while the vertical axis represents the context length in words. Results are presented for cache budgets of 256, 128, and 64. Notably, IceCache achieves 100% retrieval accuracy across all tested budget sizes.

Table 1: Accuracy comparison of our method (ICE) with SnapKV (SKV), SteamingLLM (SLM), OmniKV (OKV), MagicPig (MPG), PQCache (PQC), ArkVale (AKV), Full KV (FULL) and ground-truth top-$k$ (TOP-$k$) on LongBench for Llama-3.1-8B-Instruct and Mistral-7B-Instruct. IceCache generally outperforms other methods across various KV-cache budgets and LLMs.

| Budget | Method | Single-Document QA | | | Multi-Document QA | | | Summarization | | | Few-shot Learning | | | Synthetic | | Code | | Avg. |
|---|---|---|---|---|---|---|---|---|---|---|---|---|---|---|---|---|---|---|
| | | NrtvQA 18409 | Qasper 3619 | MF-en 4559 | HotpotQA 9151 | 2WikiMQA 4887 | Musique 11214 | GovReport 8734 | QMSum 10614 | MultiNews 2113 | TREC 5177 | TriviaQA 8209 | SAMSum 6258 | PCount 11141 | PRe 9289 | Lcc 1235 | RB-P 4206 | |
| | | | | | | | | Llama-3.1-8B-Instruct | | | | | | | | | | |
| N/A | FULL | 30.2 | 45.5 | 54.9 | 55.5 | 46.7 | 31.3 | 35.2 | 25.2 | 27.2 | 72.5 | 91.7 | 43.8 | 8.4 | 99.5 | 65.1 | 58.8 | 49.5 |
| 256 | TOP-$k$ | 30.7 | 44.7 | 55.4 | 55.0 | 46.5 | 31.7 | 34.8 | 25.1 | 26.8 | 71.5 | 92.2 | 44.8 | 7.8 | 100.0 | 67.1 | 63.6 | 49.8 |
| 256 | SKV | 23.7 | 27.3 | 46.3 | 52.3 | 40.8 | 24.2 | 19.2 | 22.6 | 19.2 | 29.0 | 84.1 | 39.0 | 8.6 | 97.5 | 57.3 | 56.2 | 40.8 |
| | SLM | 17.0 | 23.2 | 25.7 | 21.0 | 29.3 | 6.8 | 20.8 | 17.5 | 20.7 | 45.5 | 84.3 | 41.2 | 5.0 | 71.5 | 59.5 | 47.5 | 33.5 |
| | OKV | 27.2 | 40.7 | 52.8 | 55.1 | 45.6 | 29.7 | 27.6 | 23.0 | 25.4 | 72.5 | 88.9 | 40.4 | 5.6 | 94.5 | 60.8 | 51.5 | 46.3 |
| | MPG | 25.5 | 39.9 | 51.8 | 51.4 | 39.5 | 25.7 | 33.9 | 23.5 | 25.9 | 65.5 | 84.0 | 37.0 | 7.8 | 99.5 | 47.4 | 44.8 | 44.6 |
| | PQC | 28.7 | 43.3 | 52.2 | 55.2 | 45.1 | 28.4 | 27.2 | 24.0 | 22.8 | 69.5 | 91.1 | 41.2 | 6.2 | 99.0 | 59.0 | 54.4 | 47.3 |
| | AKV | 26.1 | 35.2 | 47.5 | 51.6 | 45.6 | 28.1 | 22.9 | 22.5 | 22.8 | 53.5 | 90.1 | 40.0 | 6.7 | 85.0 | 57.7 | 48.9 | 42.8 |
| 64 | ICE | 27.4 | 43.2 | 55.7 | 55.3 | 44.4 | 31.2 | 33.4 | 23.7 | 26.2 | 72.5 | 90.3 | 41.9 | 6.6 | 99.5 | 61.7 | 51.6 | 47.8 |
| 128 | ICE | 30.0 | 44.7 | 56.5 | 55.0 | 45.4 | 30.0 | 33.5 | 24.3 | 26.5 | 73.0 | 91.3 | 42.4 | 6.5 | 100.0 | 61.5 | 56.7 | 48.6 |
| 256 | ICE | 30.6 | 44.7 | 56.3 | 55.2 | 45.9 | 30.6 | 34.6 | 24.4 | 26.7 | 73.0 | 92.0 | 43.5 | 6.7 | 100.0 | 62.5 | 56.4 | 49.0 |
| | | | | | | | | Mistral-7B-Instruct-v0.2 | | | | | | | | | | |
| N/A | FULL | 26.8 | 33.1 | 49.3 | 42.8 | 27.3 | 18.8 | 33.0 | 24.2 | 27.1 | 71.0 | 86.2 | 42.8 | 2.9 | 87.0 | 56.9 | 54.3 | 42.2 |
| 256 | TOP-$k$ | 26.2 | 31.3 | 48.9 | 40.0 | 26.2 | 19.6 | 33.3 | 24.2 | 27.1 | 73.0 | 86.6 | 43.3 | 2.3 | 82.9 | 59.0 | 57.3 | 42.6 |
| 256 | SKV | 18.3 | 15.1 | 38.3 | 30.4 | 19.2 | 13.8 | 16.5 | 21.4 | 19.9 | 32.0 | 81.7 | 38.1 | 4.0 | 59.1 | 49.0 | 51.8 | 31.8 |
| | SLM | 14.2 | 12.3 | 26.4 | 23.2 | 14.8 | 10.1 | 17.5 | 19.8 | 18.8 | 51.0 | 80.5 | 39.9 | 4.0 | 15.6 | 52.0 | 45.4 | 27.8 |
| | OKV | 16.5 | 22.8 | 43.8 | 34.7 | 19.2 | 16.6 | 24.4 | 22.0 | 24.8 | 70.5 | 81.7 | 38.6 | 3.1 | 35.2 | 36.5 | 38.4 | 33.0 |
| | MPG | 21.0 | 28.0 | 46.5 | 38.4 | 19.5 | 17.5 | 30.8 | 23.3 | 25.8 | 70.0 | 83.7 | 39.5 | 2.5 | 85.0 | 49.4 | 48.3 | 39.1 |
| | PQC | 21.0 | 25.8 | 42.5 | 18.1 | 20.3 | 16.0 | 29.5 | 21.7 | 27.1 | 70.0 | 85.8 | 39.7 | 2.5 | 75.5 | 54.2 | 49.2 | 37.4 |
| | AKV | 18.0 | 15.2 | 41.5 | 30.7 | 17.0 | 13.2 | 21.6 | 21.6 | 23.1 | 58.0 | 85.8 | 40.6 | 2.1 | 41.5 | 51.4 | 47.2 | 33.0 |
| 64 | ICE | 23.9 | 29.0 | 47.4 | 40.5 | 23.5 | 18.3 | 30.6 | 21.8 | 26.0 | 70.5 | 85.9 | 41.5 | 3.5 | 56.5 | 53.2 | 50.4 | 39.0 |
| 128 | ICE | 25.1 | 30.5 | 48.7 | 40.4 | 26.7 | 18.7 | 30.3 | 22.0 | 26.1 | 70.5 | 85.7 | 42.4 | 3.4 | 70.2 | 53.6 | 51.5 | 40.4 |
| 256 | ICE | 25.2 | 31.0 | 48.4 | 40.4 | 26.0 | 18.3 | 31.5 | 23.1 | 26.9 | 71.0 | 86.3 | 42.8 | 3.9 | 85.1 | 54.7 | 53.2 | 41.7 |

PQCache, which scores 47.3 while operating with a 4× larger budget of 256. This highlights the exceptional efficiency of our approach in low-resource environments. As we increase the budget for IceCache to 256, its performance climbs to an average score of 49.0. This not only represents a substantial 1.7 point improvement over PQCache but also closes the performance gap to the unconstrained Full KV-Cache (49.5) to a mere 0.5 points. Notably, our performance is remarkably close to the ground-truth Top-$k$, demonstrating a near-optimal cache management strategy across a diverse set of tasks.

**Accuracy on Mistral-7B-Instruct.** This strong performance trend is consistent on the Mistral-7B model, confirming the robustness of our method. With a budget of 256, IceCache achieves an average accuracy of 41.7, establishing a significant 2.6 point lead over the best-performing baseline, MagicPig (39.1). Again, the low-budget capability of IceCache stands out; with a budget of 64, IceCache scores 39.0, remaining highly competitive with the top baseline (MagicPig, scores 39.1) that uses four times the cache size (256).

**Accuracy on two additional LLMs.** To further validate IceCache on larger-scale models and standard multi-head attention architectures, we evaluate it on LongBench using two additional models: Qwen3-32B and LongChat-7B-v1.5. As shown in Tables 2, for Qwen3-32B, IceCache with

Table 2: Accuracy comparison of our method (ICE) with Full KV (FULL) on LongBench for Qwen3-32B and LongChat-7B-v1.5.

| Budget | Method | Single-Document QA | | | Multi-Document QA | | | Summarization | | | Few-shot Learning | | | Synthetic | | Code | | Avg. |
|---|---|---|---|---|---|---|---|---|---|---|---|---|---|---|---|---|---|---|
| | | NrtvQA 18409 | Qasper 3619 | MF-en 4559 | HotpotQA 9151 | 2WikiMQA 4887 | Musique 11214 | GovReport 8734 | QMSum 10614 | MultiNews 2113 | TREC 5177 | TriviaQA 8209 | SAMSum 6258 | PCount 11141 | PRe 9289 | Lcc 1235 | RB-P 4206 | |
| | | | | | | | | **Qwen3-32B** | | | | | | | | | | |
| N/A | FULL | 32.6 | 45.5 | 50.4 | 59.6 | 56.0 | 40.1 | 33.2 | 23.9 | 25.2 | 72.0 | 70.8 | 37.1 | 18.0 | 100.0 | 12.4 | 18.4 | 43.4 |
| 64 | ICE | 29.5 | 43.9 | 50.9 | 61.4 | 55.8 | 38.8 | 30.8 | 23.8 | 24.3 | 71.0 | 70.0 | 35.7 | 15.0 | 97.0 | 10.5 | 17.3 | 42.2 |
| 128 | ICE | 32.1 | 44.8 | 53.3 | 61.6 | 54.6 | 38.5 | 31.0 | 23.6 | 24.8 | 72.0 | 70.3 | 36.2 | 15.5 | 100.0 | 10.7 | 17.3 | 42.6 |
| 256 | ICE | 32.2 | 44.1 | 51.9 | 60.2 | 55.1 | 38.9 | 32.4 | 24.3 | 24.7 | 71.5 | 70.7 | 37.4 | 16.5 | 100.0 | 11.8 | 18.0 | 43.1 |
| | | | | | | | | **LongChat-7B-v1.5** | | | | | | | | | | |
| N/A | FULL | 20.8 | 29.4 | 43.1 | 33.0 | 24.4 | 14.7 | 30.8 | 22.8 | 26.7 | 66.5 | 84.0 | 22.5 | 0.0 | 30.5 | 54.7 | 59.2 | 35.2 |
| 64 | ICE | 18.5 | 26.9 | 40.6 | 34.2 | 22.7 | 14.0 | 29.3 | 20.9 | 25.6 | 66.5 | 84.3 | 21.9 | 1.5 | 26.1 | 52.5 | 56.8 | 33.9 |
| 128 | ICE | 19.9 | 27.7 | 40.9 | 33.9 | 24.2 | 14.4 | 28.6 | 21.8 | 25.9 | 66.5 | 84.2 | 22.3 | 1.5 | 27.3 | 52.6 | 57.7 | 34.3 |
| 256 | ICE | 20.4 | 29.5 | 43.0 | 34.6 | 23.7 | 14.2 | 29.8 | 22.7 | 26.1 | 66.5 | 84.7 | 22.9 | 0.0 | 28.5 | 53.6 | 59.0 | 35.0 |

a small budget of 64 achieves an average accuracy of 42.2 on LongBench, retaining 97.2% of the full KV-cache performance (43.4). This score rises to 99.3% with a budget of 256, nearly matching the vanilla model. Similarly, on LongChat-7B-v1.5, our method preserves 96.3% of the full KV-cache performance with a budget of 64, and achieves up to 99.4% at a budget of 256. These results provide strong evidence that IceCache is effective across different model scales and attention mechanisms.

### 5.3.2 LATENCY ANALYSIS

Since most baselines, including IceCache, use the entire KV-cache to generate the first token, we follow PQCache (Zhang et al., 2024a) and report the Time to the second token (TT2T) in Fig. 7a, for a 36k sequence length with Llama-3.1-8B-Instruct and all the methods compared in Table 1, except MagicPig, which is restricted to costly AVX-512 CPUs. Additionally, we present results for IceCache incorporating the critical-page-reuse technique (reusing the same selected KV-page indices across layers), a method also used in OmniKV, which serves as an approximate version that trades accuracy for increased speed.

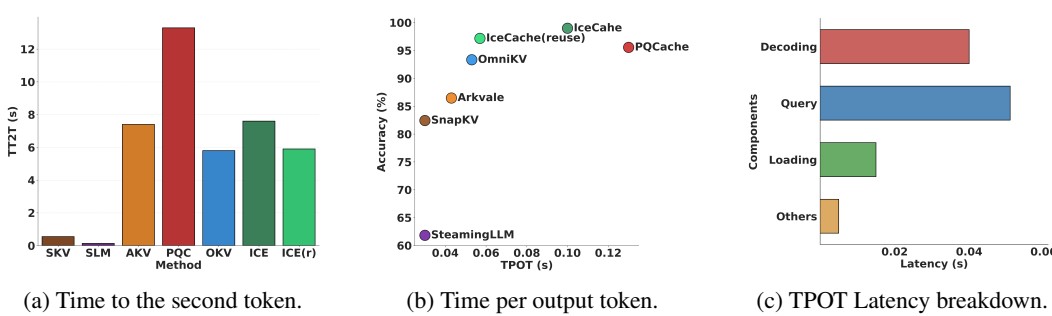

(a) Time to the second token.     (b) Time per output token.     (c) TPOT Latency breakdown.

Figure 7: Latency comparison of IceCache and baseline methods on a 36k-token sequence.

As illustrated in the figure, our method, IceCache, achieves competitive latency among retrieval-based algorithms, recording 7.7 seconds. IceCache (reuse) further reduces this to 5.9 seconds, matching OmniKV (5.8 s) and outperforming other retrieval-based baselines such as Arkvale (7.4 s) and PQCache (13.3 s). Although eviction-based methods like SnapKV (0.55 s) and StreamingLLM (0.13 s) are much faster, their speed often comes at the cost of accuracy. Overall, IceCache and IceCache(reuse) offer a strong balance between efficiency and accuracy, with the reuse variant showing how our approach can further optimize latency without significantly sacrificing performance. We include more details of IceCache(reuse) in the appendix C.

Similarly, for decoding latency, Fig. 7b shows the average time per generated token, along with the corresponding accuracy percentage relative to the full KV-cache model, for an input sequence length of 36k. Eviction-based methods, StreamingLLM and SnapKV (both at 0.03 seconds per token), continue to show the fastest speeds due to their minimal overhead. Among the more accurate retrieval-based methods, IceCache(reuse) achieves a highly competitive decoding time of 0.06 seconds per token – substantially faster than PQCache (0.13 s) and nearly matching the speed of OmniKV (0.05 s). Vanilla IceCache maintains a strong balance, combining superior accuracy with efficient decoding.

It achieves the highest accuracy percentage (99.0%) while still outperforming PQCache in speed. These results further demonstrate that IceCache effectively balances accuracy and decoding latency.

Figure 7c presents a detailed breakdown of TPOT latency for IceCache at a sequence length of 36k, with a total latency of 0.11 seconds. In this figure, "Loading", "Query", and "Decoding" correspond to the overhead from CPU–GPU communication, DCI-query operations, and the overall LLM decoding process, respectively. The largest contributors to latency are the DCI-query module (0.05 s) and decoding (0.04 s), while GPU–CPU offloading and other miscellaneous operations add only 0.015 seconds and 0.005 seconds, respectively.

### 5.4 GSM8K CoT REASONING

We also evaluate IceCache on the GSM8K benchmark using Chain-of-Thought prompting, applying a 10% budget for all compared methods using Mistral-7B-Instruct-v0.2. As shown in Table 3, Ice-Cache demonstrates superior performance, achieving an accuracy of 47.4%, significantly higher than all other methods under the same budget constraint. In particular, it improves upon the strongest baseline, PQCache (46%), by 1.2% absolute points, reaching 47.4%. Moreover, our approach nearly matches the full KV-cache (48.2%), highlighting the effectiveness of IceCache.

Table 3: GSM8K CoT.

| Method | Budget | Accuracy |
|--------|--------|----------|
| FULL | N/A | 48.2 |
| SKV | 10% | 44.7 |
| SLM | 10% | 44.4 |
| OKV | 10% | 42.7 |
| MPG | 10% | 43.1 |
| PQC | 10% | 46.2 |
| AKV | 10% | 30.9 |
| ICE | 10% | **47.4** |

### 5.5 SCALABILITY TO EXTREMELY LONG CONTEXT

To further evaluate the performance of IceCache under extremely long-context settings, we conduct experiments on the RULER benchmark and report accuracy on Single-Needle-in-a-Haystack (Single-NIAH), Multi-keys NIAH, and QA tasks with context lengths of 150k, 200k, and 250k tokens, using a token budget of 256. The experiments are conducted using Qwen3-4B-Instruct-2507 on a single H100 GPU with 64 CPU threads enabled. As shown in Table 4, IceCache and IceCache(r) consistently maintain accuracy comparable to Full-KV across all tasks and context lengths. Figure 8 further demonstrates that, as the input context grows, both IceCache and IceCache(r) exhibit a substantially slower increase in per-token decoding latency than Full-KV, whose decoding cost scales sharply with sequence length. These results indicate that IceCache achieves a better accuracy–latency trade-off for extremely long-context inference, enabling scalable decoding without sacrificing accuracy on the RULER benchmark.

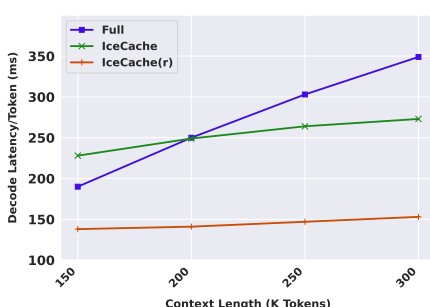

Figure 8: Latency scaling across context lengths (150k, 200k, 250k and 300k).

Table 4: Accuracy comparison on Qwen3-4B-Instruct under different context lengths (150k, 200k, 250k) on RULER benchmark.

| Method | Single-NIAH | Multi-keys NIAH | QA |
|--------|-------------|-----------------|-----|
| **150k Tokens** | | | |
| Full KV | 100.0 | 97.0 | 93.0 |
| IceCache | 100.0 | 98.0 | 92.0 |
| IceCache (r) | 100.0 | 98.0 | 91.0 |
| **200k Tokens** | | | |
| Full KV | 100.0 | 92.0 | 95.0 |
| IceCache | 100.0 | 91.3 | 95.0 |
| IceCache (r) | 100.0 | 91.0 | 94.3 |
| **250k Tokens** | | | |
| Full KV | 100.0 | 91.0 | 91.3 |
| IceCache | 100.0 | 93.0 | 91.7 |
| IceCache (r) | 100.0 | 92.0 | 92.0 |

## 6 CONCLUSION

This paper addresses the fundamental challenge of long-context inference in LLMs, where the rapidly growing KV-cache consumes substantial GPU memory and degrades computational efficiency. We first introduce the DCI-tree, a hierarchical indexing structure that integrates naturally with PagedAttention and supports dynamic updates for efficient KV-cache organization. Building on this foundation, we propose IceCache, a page-based KV-cache management framework that combines semantic token clustering with efficient GPU–CPU offloading. Across diverse long-context benchmarks, IceCache consistently achieves superior accuracy–latency trade-offs. These results establish IceCache as a scalable and practical solution for memory-efficient long-context LLM inference.

## 7    ACKNOWLEDGEMENTS

We thank the anonymous reviewers for their insightful feedback and constructive suggestions. This research was enabled in part by support provided by NSERC, the BC DRI Group and the Digital Research Alliance of Canada.

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

## A  METHOD OVERVIEW

We separate all tokens into three groups: *sink tokens*, which are the tokens at the very beginning of the input sequence; *window tokens*, which are the most recent tokens; and all the remaining tokens in between. The pages that store sink tokens are referred to as *sink pages*, and those that store window tokens are referred to as *window pages*. We always keep all the sink and window pages in GPU.

We provide the pseudocode below for IceCache. It operates in two main phases: (1) Prefill Phase: During the initial processing of prompt tokens, IceCache allocates paged KV memory per layer and performs self-attention computations. From the third layer onward, it copies KV embeddings to CPU and builds a dynamic index – DCI-tree. This tree enables efficient future lookup of important tokens based on query embeddings. (2) Decode Phase: During autoregressive decoding, each new token's query embedding is used to retrieve the most relevant KV pages via DCI-based query. Selected pages are back-loaded to GPU on demand, while unimportant pages are offloaded to CPU storage. When a new window page is offloaded, the DCI-tree is incrementally updated to store tokens in this page. The detailed steps are outlined in Algorithm 1. We will explain the mechanisms behind indexing and page selection in the following sections.

## B  METHOD DETAILS

### B.1  INDEXING

For each attention head, given a set of pre-computed key embeddings, IceCache first indexes them using a hierarchical tree structure which is obtained by a novel approach called Multi-level DCI (M-DCI). It works by constructing a dynamic index called DCI-tree and applies Prioritized DCI (P-DCI) (Li & Malik, 2017) to each level of the tree recursively (more details are in Section 3.3). The data points processed in M-DCI are transformed key embeddings and query embeddings using the following transformation formulas, which we denote as $T_K : \mathbb{R}^d \to \mathbb{R}^{d+1}$ and $T_Q : \mathbb{R}^d \to \mathbb{R}^{d+1}$:

$$T_K(\mathbf{k}_j) = \begin{bmatrix} \mathbf{k}_j/c & \sqrt{1 - \|\mathbf{k}_j\|_2^2/c^2} \end{bmatrix}^\top \tag{2}$$

$$T_Q(\mathbf{q}_i) = \begin{bmatrix} \mathbf{q}_i/\|\mathbf{q}_i\|_2 & 0 \end{bmatrix}^\top \tag{3}$$

where $c \geq \max_{j'} \|\mathbf{k}_{j'}\|_2$ is at least the maximum norm across all keys. We use the Euclidean distance as the distance function.

At the very beginning of the indexing, all data points are initially placed at the bottom level of the DCI-tree. Subsequently, some points are randomly selected to be promoted to the next higher level based on a promotion ratio $r < 1$. The ratio $r$ is predefined during DCI-tree initialization and remains fixed throughout the process. After the indexing, we can get the total number of levels in the DCI-tree, denoted as $L$. The details are presented in Algorithm 2.

Specifically, let $n_\ell$ denote the number of data points at level $\ell$, with level indices starting from the bottom (i.e., the lowest level is $\ell = 1$). Ideally, the number of points satisfies the recurrence relation $n_\ell = r \cdot n_{\ell-1}$. In other words, the distribution over level indices follows a geometric distribution. The probability that a point is assigned to the highest level ($\ell = L$) is $r^{L-1}$, while the probability of being assigned to level $\ell$ (for $1 \leq \ell \leq L - 1$) is $r^{\ell-1} - r^\ell$.

After level assignment, each data point at level $\ell$ is linked to a parent at level $\ell + 1$, defined as the closest point in terms of key embedding distance. This parent assignment is formulated as a 1-nearest neighbor search and is efficiently solved using M-DCI query.

In the decoding stage, when a new token is generated, its key embedding is inserted into the appropriate position in the DCI-tree. A level $\ell$ is first assigned to the new key according to the same random promotion process. Then, its parent at level $\ell + 1$ is determined, and the key is added to the physical memory page corresponding to the node into which it is inserted.

### B.2  PAGE SELECTION

As aforementioned, IceCache aims to accelerate self-attention by loading only a limited number of pages into GPU memory for computation. Therefore, the objective of page selection is to maximize

the *recall* (or hit rate) of significant keys for a given query. By clustering semantically similar tokens into the same nodes/pages, IceCache enables more targeted and efficient retrieval during decoding. In contrast, methods like Quest (Tang et al., 2024), Arkvale (Chen et al., 2024a), or PQCache (Zhang et al., 2024a) construct pages based on the original token order, which often causes tokens relevant to a given query to be scattered across multiple pages. Retrieving them requires loading entire pages filled with many irrelevant tokens, resulting in unnecessary memory overhead. IceCache mitigates this inefficiency by grouping similar tokens, so relevant tokens tend to be concentrated within fewer pages. As a result, the hit rate of significant keys during decoding increases. The detailed procedure is shown in Algorithms 3 and 4.

Specifically, when computing the attention matrix, given a query vector $q_i$, we follow the query process described in Section 3.3 to identify the top-$k$ keys that are most likely to yield the highest dot-product values with $q_i$. Once these top-$k$ keys are identified, we load only the pages that contain them. Suppose $p$ pages are loaded, and each page contains $d$ entries, since not all the pages are fully filled, the number of loaded keys $N$ is bounded as: $N \leq pd$.

The approximate attention scores between the query $q$ and these $N$ selected keys are then computed using Equation 1. The masks $s_{i,j}$ are set to 1 for the selected keys and 0 for all others. Note that, IceCache constructs a separate DCI-tree for each attention head, allowing it to retrieve different sets of significant pages per head. This head-specific, fine-grained selection mechanism distinguishes IceCache from baselines such as Quest and ArkVale, which retrieve the same set of pages for all heads, potentially limiting their retrieval accuracy.

## C DETAILS OF ICECACHE (REUSE) ON LONGBENCH

We present the LongBench scores for IceCache (reuse) in Table 5. Starting from the third layer, we build the DCI-tree and perform DCI-queries every three layers; we refer to these as "anchor layers". For the layers in between, we reuse the KV-cache indices selected at the most recent anchor layer.

Table 5: Accuracy of IceCache (reuse) on LongBench.

| Budget | Method | Single-Document QA | | | Multi-Document QA | | | Summarization | | | Few-shot Learning | | | Synthetic | | Code | | Avg. |
|---|---|---|---|---|---|---|---|---|---|---|---|---|---|---|---|---|---|---|
| | | NrtvQA 18409 | Qasper 3619 | MF-en 4559 | HotpotQA 9151 | 2WikiMQA 4887 | Musique 11214 | GovReport 8734 | QMSum 10614 | MultiNews 2113 | TREC 5177 | TriviaQA 8209 | SAMSum 6258 | PCount 11141 | PRe 9289 | Lcc 1235 | RB-P 4206 | |
| Llama-3.1-8B-Instruct | | | | | | | | | | | | | | | | | | |
| 256 | ICE(r) | 29.1 | 43.6 | 53.4 | 55.1 | 44.3 | 29.0 | 31.8 | 24.3 | 26.2 | 72.0 | 91.1 | 41.0 | 7.2 | 100.0 | 60.7 | 53.1 | 47.7 |

## D PERFORMANCE ON LONG GENERATION BENCHMARK

To evaluate the effectiveness of IceCache on long-context generation tasks, we conduct experiments on LongGenBench (Wu et al., 2024) using Llama-3.1-8B-Instruct with a 256-token budget. As shown in Table 6, IceCache substantially outperforms PQCache while maintaining accuracies on par with the Full-KV baseline.

Table 6: Accuracy comparison on LongGenBench for Llama-3.1-8B-Instruct.

| Method | Completion Rate | Accuracy Once | Accuracy Range | Accuracy Periodic | Avg. Accuracy |
|---|---|---|---|---|---|
| Full KV | 97.627 | 0.349 | 0.488 | 0.135 | 0.324 |
| IceCache | 95.322 | 0.377 | 0.454 | 0.164 | 0.331 |
| PQCache | 88.691 | 0.318 | 0.395 | 0.105 | 0.273 |

## E LLM USAGE

This work focuses on optimizing KV-cache management for large language models (LLMs). All the base models used in this paper can be viewed as LLMs, including Llama-3.1-8B-Instruct, Mistral-7B-Instruct-v0.2, Qwen3-32B, LongChat-7B-v1.5 and Qwen3-4B-Instruct-2507. We also leveraged an LLM to help refine the manuscript's language and improve its overall readability.

---

**Algorithm 1** IceCache

---

1: **Input:** Sequence of tokens $x_{1:I}$, Transformer with $L$ attention layers, Page size $s$
2: **Phase 1: Prefill**
3: **for** $\ell = 0$ to $L - 1$ **do**
4:     Allocate pages and arrange KVs to the pages for layer $\ell$
5:     **if** $\ell \geq 2$ **then**
6:         Copy KVs of tokens between sink tokens and window tokens from GPU to CPU (denoted as $S_k$ and $S_v$)
7:     **end if**
8:     Compute the output from the current self-attention layer $\ell$
9:     **if** $\ell \geq 2$ **then**
10:         $T_l \leftarrow$ DCI-INDEXING($S_k, S_v$)
11:     **end if**
12: **end for**
13: **Phase 2: Decode (repeated over time steps $i > I$)**
14: **while** receive new token $x_i$ with $\mathbf{q}_i$ as its query embedding **do**
15:     **for** $\ell = 0$ to $L - 1$ **do**
16:         **if** Number of tokens in the last page $\geq s - 1$ **then**
17:             Offload the oldest window page $P_w$ from GPU to CPU
18:             Set Flag to True
19:         **end if**
20:         Append KVs of $x_i$ to the end of the newest window page
21:         **if** $\ell \geq 2$ **then**
22:             $S_l \leftarrow$ PAGE-SELECT($\mathbf{q}_i, T_l, k$)
23:             Recall selected pages $S_l$ from CPU to GPU
24:         **end if**
25:         Compute the output from the current self-attention layer $\ell$
26:         **if** $\ell \geq 2$ & Flag is True **then**
27:             // Insert the tokens in offloaded $P_w$ to $T_l$
28:             **for** $i$ in $P_w$ **do**
29:                 // Random Promotion
30:                 $l_i \leftarrow 1$                         ▷ Start at bottom level
31:                 **while** Random$(0, 1) < r$ **do**
32:                     $l_i \leftarrow l_i + 1$                     ▷ Promote to the next higher level
33:                 **end while**
34:                 $\{p_i\} \leftarrow$ QUERY($\mathbf{k}_i, T_l, l_i, 1$)     ▷ Get the parent index using QUERY (Alg.4)
35:                 Insert $\mathbf{k}_i$ to $T_l$ with $p_i$ as its parent node index
36:             **end for**
37:         **end if**
38:     **end for**
39:     **if** $x_i = $ EOS **then**
40:         Break
41:     **end if**
42:     $i = i + 1$
43: **end while**

---

---

**Algorithm 2** Indexing

---

**Require:** A list $S_k$ of $n$ keys $\mathbf{k}_1, \dots, \mathbf{k}_n \in \mathbb{R}^d$, Promotion ratio $r$
    **function** DCI-INDEXING($S_k, r$)
        **for** $i = 1$ **to** $n$ **do**
            // Random Promotion
            $l_i \leftarrow 1$                                          ▷ Start at bottom level
            **while** Random$(0, 1) < r$ **do**
                $l_i \leftarrow l_i + 1$                          ▷ Promote to the next higher level
            **end while**
        **end for**
        Remove the empty levels
        Initialize $T$ with an empty root node
        **for** $i = 1$ **to** $n$ **do**
            $\{p_i\} \leftarrow$ QUERY($\mathbf{k}_i, T, l_i, 1$)            ▷ Get the parent index using QUERY (Alg.4)
            Insert $\mathbf{k}_i$ to $T$ with $p_i$ as its parent node index
        **end for**
        **return** $T$
    **end function**

---

**Algorithm 3** Page Selection

---

**Require:** Query vector $\mathbf{q}_i \in \mathbb{R}^d$, DCI-tree $T$, Number of critical keys $k$
    **function** PAGE-SELECT($\mathbf{q}_i, T, k$)
        Initialize $S_l \leftarrow \emptyset$
        $S_k \leftarrow$ QUERY($\mathbf{q}_i, T, -1, k$)
        $S_l \leftarrow$ FIND-PAGE-INDEX($S_k$)
        **return** $S_l$
    **end function**

---

**Algorithm 4** $k$-Nearest Neighbour Querying

---

**Require:** Query vector $\mathbf{q}_i \in \mathbb{R}^d$, DCI-tree $T$ with L levels, Target level $l$, Number of critical keys $k$
    **function** QUERY($\mathbf{q}_i, T, l, k$)
        **if** $l = -1$ **then**
            $l \leftarrow 1$
            Set Flag to True
        **else**
            Set Flag to False
        **end if**
        $S \leftarrow \emptyset$
        $P \leftarrow$ empty priority queue with size $k$
        **for** $i = L$ **to** $l$ **do**
            $S' \leftarrow \emptyset$
            **if** $i = L$ **then**
                $S \leftarrow \{$root node$\}$
            **end if**
            **for** $s$ **in** $S$ **do**
                $S'' \leftarrow$ Prioritized-DCI-Query($\mathbf{q}_i, s, k$)
                $S' \leftarrow S' \cup S''$
            **end for**
            **if** Flag is True **or** $i = l$ **then**
                **for** $s$ **in** $S'$ **do**
                    $P \leftarrow$ Add-to-Priority-Queue ($P, s$)
                **end for**
            **end if**
            $S \leftarrow S'$
        **end for**
        **return** $k$ nodes in $P$ that have the keys with maximum inner-product with $\mathbf{q}_i$
    **end function**

---

