# OpenReview forum: "IceCache: Memory-Efficient KV-cache Management for Long-Sequence LLMs"
_ICLR.cc/2026/Conference — ICLR 2026 Poster_

### Official Review · Reviewer_815Q · 2025-10-21

**Soundness:** 2
**Presentation:** 3
**Contribution:** 3
**Rating:** 6
**Confidence:** 4

**Summary:**

This paper introduces IceCache, a novel KV-cache management strategy designed to mitigate memory bottlenecks for LLMs processing long sequences. The core idea is to move beyond storing the KV-cache in its original token order. Instead, IceCache clusters tokens based on the semantic similarity of their key embeddings into a hierarchical structure called a DCI-tree. During inference, it employs a fast Approximate Nearest Neighbor (ANN) search algorithm (M-DCI) on the CPU to perform a query-aware, head-specific selection of the most relevant memory pages. These selected pages are then transferred from CPU to GPU for attention computation. The system is optimized with a pipelined workflow that overlaps the CPU-based page selection with GPU computations to hide latency. Experimental results across several benchmarks and models show that IceCache can maintain high accuracy (over 99% in some cases) with a significantly reduced token budget compared to the full KV-cache and outperforms other baseline methods.

**Strengths:**

- The authors compare IceCache against a strong and sufficient set of six recent state-of-the-art KV-cache optimization methods.

- The paper is well-organized and clearly written. The motivation is well-established, the proposed method is described logically with helpful diagrams (e.g., Figure 1 and 2).

- The application of a hierarchical ANN algorithm (M-DCI) to cluster and retrieve from the high-dimensional key-embedding space of the KV-cache is a novel and interesting approach.

**Weaknesses:**

- The paper does not provide an analysis of how different parameter choices of M-DCI affect the final model accuracy and inference latency. Furthermore, the specific parameter values used in the experiments are not clearly described, which could hinder reproducibility.

- The latency analysis in Section 5.5 focuses on "Time to the second token" (TT2T) and "Time per output token" (TPOT). While informative, it lacks a direct end-to-end runtime comparison against full attention.

- A central claim is that grouping semantically similar tokens into the same memory pages improves efficiency. However, there is no ablation study to isolate and quantify this specific contribution. Since the index selection happens on the CPU, a key question arises: does this memory layout truly enhance efficiency, or is the performance bottleneck dominated by the CPU search and the CPU-GPU data transfer overhead?

**Questions:**

1.  Could you provide performance curves showing how the inference speed (e.g., tokens per second) scales with increasing context length?

2.  The ANN search is currently performed on the CPU to overlap with GPU computations. Have you considered the feasibility of further accelerating the M-DCI query step by implementing it on the GPU?

3.  The proposed method seems to be tightly integrated with a CPU offloading strategy. Is it possible to adapt IceCache for a non-offload scenario (i.e., where the full KV-cache fits in GPU memory, but sparsity is desired to accelerate computation)? Or does the method fundamentally rely on the CPU having access to the cache for the DCI-tree query to work?

---

> ### Author Response · Authors · 2025-11-21
> **Part 1/N**
>
> We thank the reviewer for the review. Please find our responses as follows.
>
> **W1: The paper does not provide an analysis of how different parameter choices of M-DCI affect the final model accuracy and inference latency. Furthermore, the specific parameter values used in the experiments are not clearly described, which could hinder reproducibility.**
>
> A1: We provide an ablation study on the main M-DCI parameters: index depth and page size, and report their impact on accuracy and prefill/decoding latency. We found that IceCache is not sensitive to either index depth or page size. We conduct the evaluation on Llama-3.1-8B-Instruct with sequence length=36k and budget size=256. Prefill/decoding latency and the average accuracy on LongBench are reported.
>
> | Setting                       | Avg. Acc | Prefill Latency | Decoding Latency |
> |------------------------------|:--------:|:----------------------:|:-----------------------:|
> | index-depth = 4              |  48.7   |         7.3s           |         0.07s           |
> | index-depth = 5 *(used in paper)* | 49.0 |         7.8s           |         0.06s           |
> | index-depth = 6              |  48.9    |         8.1s           |         0.06s           |
>
> Model accuracy varies by less than 0.3 points across depths, and decoding latency remains almost unchanged. Prefill latency increases modestly with deeper trees. Depth=5 provides the best overall trade-off and is used in all experiments.
>
> | Setting                           | Avg. Acc | Prefill Latency | Decoding Latency  |
> |-----------------------------------|:--------:|:----------------------:|:-----------------------:|
> | Page size = 8                     |   49.0   |         7.9s           |         0.06s           |
> | Page size = 16 *(used in paper)* | 49.0|         7.8s           |         0.06s           |
> | Page size = 32                    |   48.9   |         7.7s           |         0.06s           |
>
> Accuracy and latency are highly stable across different page sizes. Prefill and decoding costs vary only slightly (<= 0.2s).
>
> **W2 & Q1: The latency analysis in Section 5.5 focuses on "Time to the second token" (TT2T) and "Time per output token" (TPOT). While informative, it lacks a direct end-to-end runtime comparison against full attention. Could you provide performance curves showing how the inference speed (e.g., tokens per second) scales with increasing context length?**
>
> A2: To provide a direct end-to-end comparison with full attention (W2) and to illustrate how inference speed scales with context length (Q1), we have generated the following performance curves (shown in Appendix D, Figure 7 of the updated PDF). These plots report both end-to-end prefill latency and per-token decoding latency for Full-KV, IceCache, and IceCache(r) across long contexts up to 128k tokens. The results show that IceCache introduces only a modest overhead during prefill, while achieving better scaling than Full Attention in the decoding stage.

---

> ### Author Response · Authors · 2025-11-21
> **Part 2/2**
>
> **W3: A central claim is that grouping semantically similar tokens into the same memory pages improves efficiency. However, there is no ablation study to isolate and quantify this specific contribution. Since the index selection happens on the CPU, a key question arises: does this memory layout truly enhance efficiency, or is the performance bottleneck dominated by the CPU search and the CPU-GPU data transfer overhead?**
>
> A3: We clarify that “grouping semantically similar tokens into the same memory pages” is not a source of additional overhead. Instead, it reduces both CPU search cost and CPU-GPU data transfer because it concentrates important tokens into fewer pages.
>
> When similar tokens are grouped into the same page, the model can retrieve the important Top-k tokens using fewer candidate pages since they are more concentrated. Fewer pages directly imply:
> 1. Less CPU–GPU data transfer (since only a small number of pages need to be fetched back to GPU).
> 2. Smaller search scope for DCI-query, because the effective “k” for top-k search becomes smaller when tokens are better clustered.
>
> These two factors jointly reduce the overall runtime.
>
> To isolate the effect of the memory layout, we compare IceCache to a version that does not cluster by semantics and instead uses the original token order for paging – ArkVale. As shown in Table 2 of the paper, ArkVale with a budget of 256 tokens performs worse than IceCache with a budget of only 64. Given a page size of 8, ArkVale therefore needs at least 32 pages to match the accuracy achieved by IceCache with only 8 pages. Below, we report DCI-query latency for 36k tokens when searching over 32, 16, and 8 important pages:
>
> | Method     | 32 Pages | 16 Pages | 8 Pages |
> |------------|:--------:|:--------:|:-------:|
> | DCI-query  | 1.82 ms  | 1.53 ms  | 1.30 ms |
>
> It is apparent that 8 pages lead to the smallest CPU–GPU data transfer cost. Therefore, both CPU search and CPU–GPU transfer increase substantially when more pages are needed. This demonstrates that semantic clustering is the reason IceCache can use far fewer pages without sacrificing accuracy, leading directly to lower search cost and reduced transfer overhead.
>
> **Q2: The ANN search is currently performed on the CPU to overlap with GPU computations. Have you considered the feasibility of further accelerating the M-DCI query step by implementing it on the GPU?**
>
> A4: We have also considered accelerating the M-DCI query on the GPU. We would need to acquire the necessary CUDA expertise to implement it on the GPU, but we agree it is an interesting direction for future work.
>
> **Q3: The proposed method seems to be tightly integrated with a CPU offloading strategy. Is it possible to adapt IceCache for a non-offload scenario (i.e., where the full KV-cache fits in GPU memory, but sparsity is desired to accelerate computation)? Or does the method fundamentally rely on the CPU having access to the cache for the DCI-tree query to work?**
>
> A5: IceCache does not fundamentally rely on CPU offloading. The core idea, that is, organizing keys into semantic pages and using M-DCI for approximate top-k selection, can be adapted to a setting where all KV caches remain on the GPU. In this case, the M-DCI search can be computed either on GPU or on CPU with GPU-resident KV pages, depending on the hardware budget. The idea can be relevant to the previous question, and we view this as a promising direction for future work.

---

> > ### Comment · Reviewer_815Q · 2025-11-24
> >
> > Thank you for your detailed rebuttal. After considering the additional information you provided, I have decided to maintain the current positive score.

---

> > > ### Author Response · Authors · 2025-11-24
> > >
> > > Thank you for taking the time to review our work and for maintaining the positive score. We appreciate your careful consideration and are glad our clarifications were helpful.

---

> ### Comment · Area_Chair_3vrh · 2025-11-23
>
> Dear reviewer 815Q,
>
> Thanks for your time and effort in reviewing ICLR2026 submissions. The authors have submitted their responses to your review. Please take the time to read and raise your further comments, and discuss with the authors. Thanks very much!
>
> AC

---

### Official Review · Reviewer_Egcx · 2025-10-25

**Soundness:** 3
**Presentation:** 2
**Contribution:** 3
**Rating:** 4
**Confidence:** 4

**Summary:**

IceCache introduces a new approach to memory-efficient KV-cache management for long-sequence LLMs. It uses a hierarchical DCI-tree that clusters key embeddings based on semantic similarity, grouping related tokens into memory pages. IceCache further employs bulk data loading and CPU-GPU pipelining to minimize latency. Tested on models like LLama-3.1-8B, Mistral-7B, Qwen3-32B, LongChat-7B on various tasks. It maintains over 99% accuracy with a 256 token budget.

**Strengths:**

1. High efficiency and accuracy: while cutting KV memory usage, achieve high accuracy
2. Fine-grained retrieval: per-head-query-aware selection, improve attention focus
3. Efficient pipelining: CPU-GPU overlap computation with data movement to reduce latency

**Weaknesses:**

1. Limited analysis: missing ablations on index depth, page size and computational cost of clustering and updates
2. Scalability uncertainty: effectiveness and efficiency on extremely long contexts or distributed multi-GPU settings
3. Figure clarity and presentation issues: figure 1 & 4 are not well-explained or visually clear. lack of consistent notation and labeling.

**Questions:**

1. What is the latency for prefill stage, since you introduce additional clustering and indexing
2. Did you try on multi-turn conversation where context may change, since I assume the serial workflow in Figure 4 means multi-turn. What is the latency and accuracy.
3. Figure 4 is lack of explanation and notations, hard to understand.
4. Did IceCache select page for each decoding step or only once for the whole decoding phase.

---

> ### Author Response · Authors · 2025-11-21
> **Part 1/N**
>
> We thank the reviewer for the review. Please find our responses as follows.
>
> **W1: Limited analysis: missing ablations on index depth, page size and computational cost of clustering and updates**
>
> A1: We performed additional ablations on index depth, page size, and the computational cost of clustering and updates per attention layer (performed in parallel for all attention heads). We found that the computational costs of both clustering and updates are not significantly affected by either index depth or page size. The results (Llama-3.1-8B-Instruct, seq=36k, budget=256) are summarized below.
>
> | Setting         | Clustering Latency | Update Latency |
> |-----------------|:------------------:|:---------------:|
> | index-depth = 4 |      0.053s        |    1.79 ms      |
> | index-depth = 5 |      0.055s        |    1.83 ms      |
> | index-depth = 6 |      0.058s        |    1.96 ms      |
>
> Increasing depth adds only ~5 ms of clustering cost per layer and ~0.2 ms per update, showing that IceCache is not sensitive to index depth.
>
> | Setting        | Clustering Latency | Update Latency |
> |----------------|:------------------:|:---------------:|
> | Page size = 8  |       0.059s       |     1.99 ms     |
> | Page size = 16 |       0.055s       |     1.83 ms     |
> | Page size = 32 |       0.053s       |     1.80 ms     |
>
>
> Page size has minimal impact on both clustering and update cost. All configurations remain efficient.
>
> **W2: Scalability uncertainty: effectiveness and efficiency on extremely long contexts or distributed multi-GPU settings**
>
> A2: In addition to the passkey-retrieval experiment up to 100k tokens reported in the paper, we further evaluate IceCache on extremely long contexts (150k, 200k, 250k) using tasks from the RULER benchmark (Single_NIAH, Multi-keys_NIAH, and QA). Because Llama-3.1-8B-Instruct supports a maximum context length of 130k tokens, we run these experiments on Qwen3-4B-Instruct, which supports context lengths up to 260k tokens. We found our method scales better than the Full KV in decoding.
>
> | Context Length | Method | Single_NIAH | Multi-keys_NIAH |   QA   | Prefill Latency | Decoding Latency |
> |:--------------:|-------------|:-----------:|:---------------:|:------:|:----------------:|:----------------:|
> | **150k**       | Full KV       |   100.0     |      97.0       |  93.0  |     26.1 s       |   189 ms (+0%)   |
> |                | IceCache      |   100.0     |      98.0       |  92.0  |     30.6 s       |   294 ms (+0%)   |
> |                | IceCache (r)  |   100.0     |      98.0       |  91.0  |     28.2 s       |   172 ms (+0%)   |
> | **200k**       | Full KV       |   100.0     |      92.0       |  95.0  |     43.8 s       |   251 ms (+32%)  |
> |                | IceCache      |   100.0     |      91.3       |  95.0  |     49.5 s       |   333 ms (+13%)  |
> |                | IceCache (r)  |   100.0     |      91.0       |  94.3  |     46.2 s       |   207 ms (+20%)  |
> | **250k**       | Full KV       |   100.0     |      91.0       |  91.3  |     67.6 s       |   310 ms (+64%)  |
> |                | IceCache      |   100.0     |      93.0       |  91.7  |     76.2 s       |   375 ms (+27%)  |
> |                | IceCache (r)  |   100.0     |      92.0       |  92.0  |     69.9 s       |   237 ms (+37%)  |
>
> The results show that:
> 1. Both IceCache and IceCache(r) match or closely track Full KV accuracy across all tasks, even at 250k token contexts.
> 2. Prefill latency remains close to the Full KV: IceCache and IceCache(r) incur only a small overhead compared to Full KV, demonstrating that M-DCI indexing remains efficient even in 250k-token scenarios.
> 3. Decoding latency scales better: As the context grows from 150k to 250k tokens, the decoding latency of IceCache and IceCache(r) increases by only 27% and 37%, respectively, which are far lower than the 64% increase observed with Full KV. This shows that IceCache scales more gracefully than full attention and achieves substantial speedups at extreme context lengths.
>
> **W3 & Q3: Figure clarity and presentation issues: figure 1 & 4 are not well-explained or visually clear. lack of consistent notation and labeling. Figure 4 is lack of explanation and notations, hard to understand.**
>
> A3: We thank the reviewer for pointing this out. We have updated the PDF.

---

> ### Author Response · Authors · 2025-11-21
> **Part 2/2**
>
> **Q1: What is the latency for prefill stage, since you introduce additional clustering and indexing**
>
> A4: The prefill latency reported in Figure 6(a) of the paper is an end-to-end measurement, and it already includes all additional overhead introduced by IceCache, such as clustering and M-DCI indexing.
>
> To provide a more explicit comparison, we also plot the end-to-end prefill latency across different context lengths (32k, 64k, 96k, 128k), shown in Appendix D, Figure 7 of the updated PDF. As the plot demonstrates, IceCache and IceCache(r) incur only a modest increase in prefill latency compared to Full-KV.
>
> **Q2: Did you try on multi-turn conversation where context may change, since I assume the serial workflow in Figure 4 means multi-turn. What is the latency and accuracy**
>
> A5: We would like to point out that Figure 4 does not refer to multi-turn conversation: (Prefill, Li) and (Prefill, Li+1) indicate the prefill computation at consecutive attention layers, not dialogue turns.  We have updated its caption in the new PDF.
>
> To evaluate multi-turn performance, we tested IceCache on the Multi-IF benchmark (https://arxiv.org/pdf/2410.15553), which is specifically designed to assess an LLM’s ability to handle evolving conversational context (Llama-3.1-8B-Instruct, budget=256). We found that compared with the eviction-based algorithm (SnapKV), IceCache can handle changed contexts much better.
>
> | Method        | Turn 1 | Turn 2 | Turn 3 |
> |---------------|:------:|:------:|:------:|
> | Full-KV       | 0.7958 | 0.7027 | 0.6305 |
> | SnapKV        | 0.8033 | 0.4864 | 0.2583 |
> | IceCache      | 0.8012 | 0.7007 | 0.6296 |
> | IceCache (r)  | 0.8001 | 0.6993 | 0.6279 |
>
> Based on these results, IceCache closely matches Full-KV accuracy across all turns, while eviction-based methods like SnapKV degrade rapidly as the context evolves. As a retrieval-based method, IceCache remains stable because it selects pages dynamically at each decoding step and continuously updates the DCI-tree, allowing important tokens to be re-identified even as new context is introduced.
>
> We extend the original multi-turn dataset to longer contexts of 32k, 64k, and 128k tokens by having the model generate additional rounds of conversation. We then measure the end-to-end decoding latency and present the results below. We found IceCache scales better than the baseline (Full-KV) in decoding.
>
> | Sequence Length | Full-KV           | IceCache          | IceCache (r)      |
> |:---------------:|:-----------------:|:-----------------:|:-----------------:|
> | 32k         | 43 ms (+0%)       | 105 ms (+0%)      | 60 ms (+0%)       |
> | 64k        | 50 ms (+16%)      | 111 ms (+6%)      | 66 ms (+10%)      |
> | 128k        | 93 ms (+116%)     | 129 ms (+23%)     | 84 ms (+40%)      |
>
> Both IceCache and IceCache (r) exhibit a much slower growth rate in the decoding latency as the sequence length increases compared to Full-KV. At very long contexts (e.g., 128k tokens), IceCache (r) becomes close or even faster than Full-KV decoding while still maintaining high accuracy and requiring substantially less GPU memory.
>
> **Q4: Did IceCache select page for each decoding step or only once for the whole decoding phase**
>
> A6: IceCache selects pages at every decoding step, not just once at the beginning of the decoding phase.

---

> ### Comment · Area_Chair_3vrh · 2025-11-23
>
> Dear reviewer Egcx,
>
> Thanks for your time and effort in reviewing ICLR2026 submissions. The authors have submitted their responses to your review. Please take the time to read and raise your further comments, and discuss with the authors. Thanks very much!
>
> AC

---

> ### Author Response · Authors · 2025-11-27
> **Kind Follow-up on Our Rebuttal**
>
> Dear reviewer Egcx,
>
> Thank you for taking the time to review our work. Kindly note that the discussion phase ends on December 2. We would be more than happy to provide additional clarification if there are any further questions or concerns.
>
> Sincerely,
>
> Authors of IceCache

---

### Official Review · Reviewer_SWPZ · 2025-10-30

**Soundness:** 3
**Presentation:** 2
**Contribution:** 3
**Rating:** 4
**Confidence:** 3

**Summary:**

This paper proposes a KV-cache management method called IceCache, combining semantic token clustering (using a hierarchical DCI-tree) with the PagedAttention mechanism to optimize long-sequence LLM inference. The core idea is to group semantically related tokens into the same memory pages, enabling more accurate and efficient query-aware page selection during decoding, compared to methods that rely on the original token order. The paper demonstrates better results compared with baselines, maintaining over 99% of full-model accuracy with a significantly reduced KV-cache budget (e.g., 256 tokens) across a wide range of long-context benchmarks.

**Strengths:**

1. Grouping tokens by semantic similarity in key-embedding space improves cache hit rates, which is powerful and clearly explained.
2. Comprehensive evaluation with diverse benchmarks and multiple models with impressive results.
3. Superior accuracy-latency trade-off compared to other high-accuracy, retrieval-based methods.

**Weaknesses:**

1. Lack quantitative analysis of the computational cost of building and maintaining the DCI-tree on the CPU. How significant is the CPU utilization? Could this become a bottleneck on a system with a less powerful CPU or when running multiple instances?
2. The storage cost of the DCI-tree indices themselves is not discussed. For a context length of 100k tokens per layer and per head, what is the memory footprint of the index on the CPU?
3. Compared to OmniKV, the overhead is primarily from the DCI-query, the more complex data movement, or both? A direct comparison of these components with a leading baseline like OmniKV would be more informative.
4. The description of M-DCI with P-DCI, but the algorithm in Page 4's pseudocode and Appendix B.1 uses a simpler promotion and parent-assignment scheme, the consistency should be improved.
5. The entire method hinges on the quality of the key embeddings for clustering. How about the assumption of the clustering not hold?
6. The promotion ratio r for the DCI-tree and the number of levels L are crucial hyperparameters. How were they chosen? How sensitive are the results to their values?

**Questions:**

The same as the weakness.

**Details Of Ethics Concerns:**

No ethics concerns.

---

> ### Author Response · Authors · 2025-11-21
> **Part 1/N**
>
> We thank the reviewer for the review. Please find our responses as follows.
>
> **W1: Lack quantitative analysis of the computational cost of building and maintaining the DCI-tree on the CPU. How significant is the CPU utilization? Could this become a bottleneck on a system with a less powerful CPU or when running multiple instances?**
>
> A1: We provide quantitative evidence that the CPU-side DCI operations are fairly lightweight. Performance does not degrade significantly with fewer threads, suggesting that it is unlikely to be a bottleneck on less powerful CPUs.
>
> 1. The CPU cost of building and maintaining the DCI-tree is small.
> Given a 36k context length, it only requires 7.5 MB of CPU memory to store the DCI-tree per attention layer and per head; in terms of the latency, DCI-tree construction takes less than 0.3 seconds per attention layer, and DCI-insert latency is only 3-4 ms per attention layer (both are negligible compared to the overall prefill and decoding latencies of a long-context model).
>
> 2. CPU utilization does not saturate and scales well with fewer CPU resources.
> To further support our point, we reduced the number of CPU threads from 64 to 32 on a long sequence (130k tokens, Llama-3.1-8B-Instruct).
> | Operation                       | #thread = 32 | #thread = 64 |
> |---------------------------------|:------------:|:------------:|
> | DCI-index per attention layer   |    0.30 s    |    0.21 s    |
> | DCI-insert per attention layer  |   0.004 s    |   0.003 s    |
>
> This modest degradation despite halving CPU parallelism demonstrates that DCI computation is not CPU-bound, and GPU-side operations remain the primary bottleneck.
>
> **W2: The storage cost of the DCI-tree indices themselves is not discussed. For a context length of 100k tokens per layer and per head, what is the memory footprint of the index on the CPU?**
>
> A2: The DCI-tree index is compact and its memory footprint scales slowly with sequence length. For Llama-3.1-8B-Instruct, we measured the memory footprint of the DCI-tree per layer and per head across long context lengths:
>
> | Seq-length | DCI-tree Memory|
> |:---------------:|:-------------------------------------:|
> | 65k         | 8.2 MB                                |
> | 130k        | 9.1 MB                                |
>
> The increase is small (<1 MB when doubling the sequence length), because the DCI structure stores only a fixed number of projections and metadata rather than full token embeddings.

---

> ### Author Response · Authors · 2025-11-21
> **Part 2/2**
>
> **W3: Compared to OmniKV, the overhead is primarily from the DCI-query, the more complex data movement, or both? A direct comparison of these components with a leading baseline like OmniKV would be more informative.**
>
> A3: Compared to OmniKV, the overhead of IceCache is primarily from the DCI-query, whereas the difference in data movement between our method and the baselines is negligible. To further analyze this question, we measure two overheads: (1) Top-k selection time and (2) data-movement time, and directly compare both components against OmniKV and PQCache.
>
> For sequence length 36k and budget size 256, the **data-movement** latency is:
> 0.019s (OmniKV), 0.024s (PQCache), 0.023s (IceCache), 0.021s (IceCache-r)
>
> These numbers are nearly identical, showing that data movement contributes minimally to the overhead differences across methods. The primary difference, therefore, comes from the Top-k selection stage.
>
> Regarding Top-k selection, we found DCI-query scales better than OmniKV, and its latency is comparable to other methods for long sequences.
>
> | Seq-length    |   36k   |     130k      |
> |:-------------:|:----------------:|:----------------:|
> | OmniKV        | 0.034 s  | 0.058 s (+70%)  |
> | PQCache       | 0.121s   | 0.174 s (+44%)  |
> | IceCache      | 0.096 s   | 0.123 s (+28%)  |
> | IceCache (r)  | 0.048 s  | 0.060 s (+25%)  |
>
> From the results, IceCache is significantly faster than PQCache, and IceCache (r) nearly matches OmniKV at 130k tokens. As the sequence length increases from 36k to 130k (3.6 times), the DCI-query latency (IceCache and IceCache-r) increases by only 28% and 25%, which is significantly smaller than OmniKV (70%) and PQCache (44%).
>
> It is difficult to compare to OmniKV directly, because it uses substantially more memory than other methods. This is because OmniKV only performs offloading for some layers and leaves all other layers fully resident on the GPU, unlike our method and other baselines. Other methods, including IceCache and PQCache, offload the KV caches of all layers after the first two layers to the CPU, and therefore must perform Top-k selection using a CPU-side approximate search algorithm. OmniKV, in contrast, keeps full KV caches for several additional middle layers directly on the GPU and performs Top-k selection only on these layers. This design significantly increases GPU-memory usage but reduces CPU-GPU data movement and enables brute-force Top-k retrieval using torch.topk() entirely on the GPU. Consequently, OmniKV has very fast Top-k selection at short sequences. However, as sequence length grows, its brute-force GPU search scales poorly. As shown earlier, OmniKV’s latency increases more rapidly than IceCache and PQCache, and beyond a certain context length, the performance ranking reverses.
>
> **W4: The description of M-DCI with P-DCI, but the algorithm in Page 4's pseudocode and Appendix B.1 uses a simpler promotion and parent-assignment scheme, the consistency should be improved.**
>
> A4: We thank the reviewer for pointing this out. We have updated the pseudocode of Algorithms 1&2 in the new PDF.
>
> **W5: The entire method hinges on the quality of the key embeddings for clustering. How about the assumption of the clustering not hold?**
>
> A5: Our method does not necessarily require all key embeddings to be well clustered. For example, if there were fewer well-clustered tokens than the number of tokens in a page, it is more advantageous to group those tokens together into the same page than otherwise. Even if the remaining tokens in that page belong to other clusters, this would still work better than placing arbitrary tokens into the same page.
>
> **W6: The promotion ratio r for the DCI-tree and the number of levels L are crucial hyperparameters. How were they chosen? How sensitive are the results to their values?**
>
> A6: We performed an ablation sweep over a range of r and L. Across all tested configurations, we found that decoding latency is not sensitive to r and L; accuracy shows moderate sensitivity, but varying only within a small range. Therefore, we chose the r and L with the best accuracy while maintaining low latency. The table below shows the decoding latency (seq-length=36k) and the average accuracy on LongBench (Llama-3.1-8B-Instruct, budget=256):
>
> | Setting                             | Avg. Acc | Decoding Latency |
> |-------------------------------------|:--------:|:----------------:|
> | r = 0.1 (corresponding to L = 4)    |  48.7   |      0.07s       |
> | r = 0.2 (corresponding to L = 5)    |  49.0   |      0.06s       |
> | r = 0.3 (corresponding to L = 6)    |  48.9   |      0.06s       |
>
>
> These results show that IceCache is stable across a broad range of r and L: accuracy varies by less than 0.3, and decoding latency is stable. The chosen configuration (r=0.2, L=5) provides the best overall accuracy-latency trade-off.

---

> ### Comment · Area_Chair_3vrh · 2025-11-23
>
> Dear reviewer SWPZ,
>
> Thanks for your time and effort in reviewing ICLR2026 submissions. The authors have submitted their responses to your review. Please take the time to read and raise your further comments, and discuss with the authors. Thanks very much!
>
> AC

---

> ### Author Response · Authors · 2025-11-27
> **Kind Follow-up on Our Rebuttal**
>
> Dear reviewer SWPZ,
>
> Thank you for taking the time to review our work. Kindly note that the discussion phase ends on December 2. We would be more than happy to provide additional clarification if there are any further questions or concerns.
>
> Sincerely,
>
> Authors of IceCache

---

### Author Response · Authors · 2025-11-21
**General Response**

We thank the reviewers for their reviews. We summarize our answers to each of the questions below. Details can be found in the response to each reviewer.

**Q1: (R SWPZ, Egcx, 815Q) Ablation study on the page size, index depth and promotion ratio.**

A1: We have added an ablation study demonstrating that IceCache remains robust under different choices of page size, index depth, and promotion ratio.

**Q2: (R SWPZ, 815Q) Analysis of the DCI-tree related operations (indexing, query, insertion).**

A2: We have reported the latency and memory footprint of the DCI-tree, showing that it is computationally efficient across different context lengths.

**Q3: (R Egcx, 815Q) End-to-end runtime (TT2T, TPOT) analysis and the comparison against full attention.**

A3: We have included Figure 7 in the new PDF that visualizes how inference latency scales with increasing context length for both IceCache and full attention, clearly demonstrating the scalability of IceCache.

**Q4: (R SWPZ) Analysis of different components of IceCache (Top-k selection, data movement).**

A4: We have reported the latency of Top-k page selection and data movement, and compared IceCache against two baselines: OmniKV and PQCache. The results show that IceCache scales better than both baselines.

**Q5: (R SWPZ) If the method is based on the assumption of good key embeddings clustering?**

A5: IceCache does not rely on a strong clustering assumption. Even when clusters are less separable, our method remains effective and does not degrade to worse behavior than arbitrary grouping.

**Q6: (R Egcx) Accuracy and latency on multi-turn conversion task.**

A6: We have added experiments on a multi-turn conversation benchmark, showing that IceCache maintains both high accuracy and low latency in that setting.

**Q7: (R Egcx) Scalability to extremely long sequences (>= 150k tokens).**

A7: We have added experiments on the RULER benchmark using 150k, 200k, and 250k context lengths. The results confirm that IceCache maintains accuracy and scalability even under extremely long contexts.

**Q8: (R 815Q) Isolate and quantify the contribution of grouping similar tokens into the same pages.**

A8: We have added the experiments and analysis showing this grouping reduces the number of selected pages without sacrificing accuracy, which in turn reduces the Top-k selection and data movement latency.

**Q9: (R 815Q) If M-DCI can be implemented on GPUs, and if it is possible to adapt IceCache for a non-offload scenario?**

A9: Yes, we have considered accelerating the M-DCI query and implementing IceCache purely on GPUs as promising future directions.

---

### Author Response · Authors · 2025-12-03
**New General Response**

We sincerely thank all reviewers for their careful and valuable feedback. We are encouraged that the reviewers consistently acknowledged IceCache’s key strengths:

1. **Novel Methodology:** The innovative application of hierarchical ANN (M-DCI) for semantic token grouping and fine-grained, per-head retrieval (all reviewers).
2. **System Efficiency:** Effective CPU-GPU pipelining that overlaps computation with data movement to minimize latency (R Egcx).
3. **Superior Performance:** IceCache achieves superior accuracy–latency and accuracy–memory trade-offs compared to leading KV-cache baselines. (R SWPZ, Egcx).
4. **Rigorous Evaluation:** A comprehensive evaluation against six SOTA baselines across diverse benchmarks (R SWPZ, 815Q).

We also appreciate the reviewers’ critical comments and questions. In response, we have substantially improved the paper in the following ways:

**Enhanced Experiments:**
*(R Egcx) Scalability to extremely long sequences (>= 150k tokens).*
*(R Egcx) Accuracy and latency on multi-turn conversion task.*
1. Added evaluation to 150k, 200k, and 250k tokens on the RULER benchmark, demonstrating IceCache scales significantly better than Full-KV.
2. Added experiments on a multi-turn conversation benchmark (Multi-IF), showing IceCache maintains stability in dynamic contexts where eviction-based methods (e.g., SnapKV) fail.

**Strengthened Analysis:**
*(R SWPZ, Egcx, 815Q) Ablation study on the page size, index depth and promotion ratio.*
*(R SWPZ, 815Q) Analysis of the DCI-tree related operations (indexing, query, insertion).*
*(R Egcx, 815Q) End-to-end runtime (TT2T, TPOT) analysis and the comparison against full attention.*
*(R SWPZ) Analysis of different components of IceCache (Top-k selection, data movement).*
*(R 815Q) Isolate and quantify the contribution of grouping similar tokens into the same pages.*
1. Added comprehensive ablations on index depth, page size, promotion ratio ($r$), and levels ($L$), proving IceCache's insensitivity to the variance of these parameters.
2. Added CPU utilization analysis and memory footprint measurements for the DCI-tree, showing that it is unlikely to be a bottleneck on less powerful CPUs and that its memory overhead is small.
3. Added end-to-end latency curves across increasing context lengths for both IceCache and Full Attention, indicating that IceCache introduces only a modest overhead during prefill, while achieving better scaling than Full Attention in the decoding stage.
4. Added direct comparisons that explicitly separate data-movement overhead and Top-k selection overhead, showing that IceCache’s efficient retrieval scales significantly better than the brute-force search used by baselines.
5. Quantified the specific benefit of semantic clustering by comparing against non-clustered paging (ArkVale).

**Improved Writing:**
*(R SWPZ) Improve the consistency between the pseudocode and the method description.*
*(R Egcx) Provide more detailed explanations for Figures 1 and 4.*
1. Updated the pseudocode (Algorithms 1 and 2) and refined the captions of Figures 1 and 4 to ensure consistency and improve clarity.

Please see the record of the rebuttal phase about how we addressed the reviewers' specific concerns in detail.

**Conclusion:**
We believe that we have addressed all major criticisms of the reviewers and kindly request that our score be increased.

---

### Meta-Review · Area_Chair_ioM1 · 2026-01-07

**Summary:**

This paper attempts to solve an interesting problem about long-context KV-cache management. The original main concerns from the reviewers were based on detailed configurations and results in the empirical studies; the authors managed to provide solid additional experimental results to resolve the concerns. Thus, I tend to recommend a marginal acceptance of the paper based on my educated guess of the reviewers' potential reply if they were given the chance.

**Reviewer Concerns:**

All the concerns about the experimental results and ablation studies have been discussed by the authors. I tend to believe that it could resolve those concerns.

**Reviewer Scores:**

I tend to believe the reviewers could increase their scores given the additional experiments.

---

### Decision · Program_Chairs · 2026-01-26

Accept (Poster)